# Satellite monitoring of terrestrial plastic waste

**Caleb Kruse**[1], **Edward Boyda**[1], **Sully Chen**[1], **Krishna Karra**[1], **Tristan Bou-Nahra**[1], **Dan Hammer**[1], **Jennifer Mathis**[2], **Taylor Maddalene**[2], **Jenna Jambeck**[2], **Fabien Laurier**[3]*

**1** Earthrise Media, Berkeley, CA, United States of America, **2** College of Engineering, University of Georgia, Athens, GA, United States of America, **3** Minderoo Foundation, Perth, Western Australia, Australia

* flaurier@minderoo.org

**Data Availability Statement:** Waste site location data and metadata about each site is accessible at https://globalplasticwatch.org/Programmatic API access to the data is available at https://api.globalplasticwatch.org/Code is public in the

## Abstract

Plastic waste is a significant environmental pollutant that is difficult to monitor. We created a system of neural networks to analyze spectral, spatial, and temporal components of Sentinel-2 satellite data to identify terrestrial aggregations of waste. The system works at wide geographic scale, finding waste sites in twelve countries across Southeast Asia. We evaluated performance in Indonesia and detected 374 waste aggregations, more than double the number of sites found in public databases. The same system deployed in Southeast Asia identifies 996 subsequently confirmed waste sites. For each detected site, we algorithmically monitor waste site footprints through time and cross-reference other datasets to generate physical and social metadata. 19% of detected waste sites are located within 200 m of a waterway. Numerous sites sit directly on riverbanks, with high risk of ocean leakage.

## Introduction

Plastics are a major pollutant impacting our planet. They are integrated into nearly all aspects of our daily life and are leaking into the environment. Plastic waste has reached the world's highest points, deepest parts of the ocean, seafloor sediment cores, populated areas, remote islands, and both poles [1–6]. On reaching the ocean, plastics persist for decades as an insidious pollutant [7, 8]. Plastics have been found to cause harm to hundreds of species, including all sea turtle species, almost half the cetacean and marine bird species, and damage coral reefs and other ecosystems [9–11]. An estimated 11 million metric tons of plastic waste currently enters the ocean each year, a rate that is expected to nearly triple by 2040 [12]. Since the sources of environmental plastics are broadly distributed geographically, most studies of pollution pathways have either focused on a selected set of field locations [13], or infer plastic waste density through modeling [14, 15]. It is more urgent than ever to actively monitor the sources of plastic pollution further upstream.

Previous research has shown that plastic in the environment and ocean is influenced by mismanaged waste on land [13, 16]. It is estimated that 70–80% of plastic pollution comes from land-based sources and that 91% of ocean plastic pollution occurs via watersheds [14]. Additionally, while there are an estimated 1,000 rivers transporting plastic waste to the ocean, the top ten are located in South or Southeast Asia where dumpsites are still commonly used for disposal [17–19].

following GitHub repository: https://github.com/earthrise-media/plastics.

**Funding:** Earthrise Media and JJ received funding from the Minderoo Foundation (https://minderoo.org/) for this work. The funders had no role in study design, data collection and analysis, decision to publish, or preparation of the manuscript.

**Competing interests:** The authors have declared that no competing interests exist.

Globally, 12% of municipal solid waste (MSW) is made up of plastics [19]. As such, identifying MSW serves as a key proxy for locating aggregations of plastics in the environment. The development of new and effective plastic management strategies requires an understanding of aggregations of municipal waste, particularly for litter hotspots, illegal dumping, and related high-risk leakage sites. Due to lack of resources to scale, government reporting can be scarce, out of date, and often doesn't account for informal waste management practices. This work seeks to leverage remote sensing data to fill this information gap, identifying waste aggregation locations in service of reducing the impacts of plastic pollution. With remote detection of waste aggregations, one can measure rather than model waste distributions and monitor waste site development through time, eventually within a globally consistent and comprehensive dataset.

To our knowledge, no operational monitoring system for plastic-bearing municipal solid waste exists. For ocean plastics, conceptual [20] and small-scale [21] studies have shown that the spectral signature of floating plastic debris is likely characterizable. Both Biermann et al. [22] and Themistocleous et al. [23] demonstrated that indices derived from multispectral Sentinel-2 data are sufficient to identify floating debris in a marine environment. On land, the spectral diversity of waste and land cover makes it challenging to devise spectral indices that can effectively discriminate waste. More recently, Gill et al. [24] developed a method for detecting large, managed landfills in Kuwait using Landsat-derived land surface temperature increases, Page et al. [25] developed a classification method for both tire and plastic waste in Scotland using Sentinel-1 and Sentinel-2 data, and a proof-of-concept system to detect municipal waste in Da Nang, Vietnam was demonstrated by the Japan Manned Space Systems Corporation and the Da Nang Institute for Socio-Economic Development [26].

Growth of computational infrastructure, architectural innovations, and new training techniques have established neural networks as preeminent systems for image classification. Recent work has shown that neural network systems have the ability to produce global datasets from Earth observation data, classifying and monitoring features at a greater level of specificity, robustness, and scale than ever before [27–30]. We build on this work, creating a novel pipeline of neural networks that parse spectral, structural, and temporal information from Sentinel-2 satellite data to identify plastic waste aggregations on land throughout Southeast Asia.

The system returns a three-fold increase in validated waste site detections over those documented on OpenStreetMap. The approach allows for a repeatable, scalable, cost-effective, and operational monitoring capability for plastic waste on land. This work is in direct support of the global observation system for marine debris as proposed by Martínez-Vicente et al. [31].

## Methods

We developed a system of neural networks to analyze spectral, spatial, and temporal characteristics of Sentinel-2 satellite data to identify sites with aggregations of waste. Supposing sufficient quantities of data and that the signal from waste is unique, one should be able to train a single convolutional neural network for the task. However, we began with only a handful of known waste site examples and found that the spectral signal of waste is varied, subtle, and noisy. Many of the resulting methods and system design features can be understood as flowing from constraints on the data, as methods to introduce additional streams of information to the neural networks.

The computational engine consists of two convolutional neural networks that analyze and combine spectral, spatial, and temporal signals. The two networks work in tandem, with candidate regions generated by the first that are then cross-validated by the second. We built the first stage of classification to operate on a per-pixel basis in order to amplify the amount of

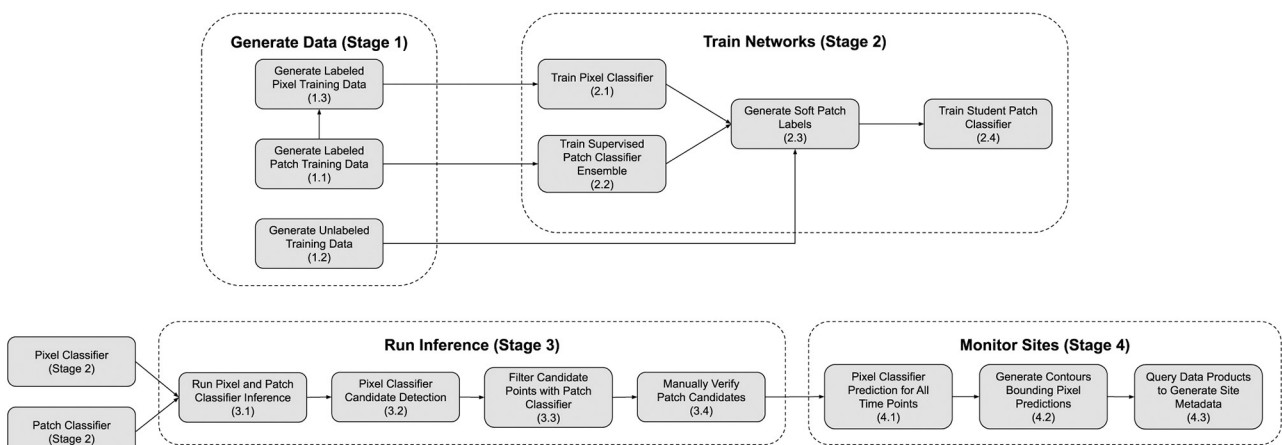

**Fig 1. The major stages of the methodological pipeline.** The components are modular, with products flowing from one stage as input to the next. There are four functional modules (data generation, network training, inference, and site monitoring). Major subcomponents of each shown and labeled for reference in the methods section. A diagram with additional detail is included in S1 Fig.

data extracted from each known site, and to limit spatial overfitting that would be seen in a classifier that incorporates spatial information. Adding a temporal component to the spectral information serves to suppress some backgrounds which share characteristics with waste site fill. For example, the turned earth of farmed fields or senescence of seasonal vegetation [32] can appear spectrally similar to a waste site, but exhibits more seasonal variation. We incorporate spatial information through a secondary patch-based classifier that validates candidates surfaced in pixel classification. Pairing these neural networks compensates for each others' biases. We augment the training dataset continuously, incorporating prior true and false positive detections, into the training dataset, and leveraging unlabeled data through semi-supervised distillation [33]. The major stages of the methodological pipeline are laid out in Fig 1 and explored in more detail through the component sections of the methods.

Unless otherwise denoted, supplied parameter values in the methods are included to establish the architecture of the detection system and to aid reproducibility.

## Data

**Data sources.**   The Copernicus Sentinel-2 program of the European Space Agency provides a globally comprehensive, open-access dataset of satellite-based Earth observations, with moderately high spatial resolution (10, 20, or 60 meters / pixel depending on the band), broad multi-spectral range (12 bands between 442 nm and 2186 nm), and frequent temporal revisit rate (5 days). Sentinel-2 data has been collected continuously since late 2015. High resolution basemap data (Google Earth, Mapbox, Bing, 30–50 cm / pixel) has proved valuable for site validation, but using the underlying proprietary imagery for detection would involve significant tradeoffs in cost, spectral range, revisit rate, and data standardization and accessibility (Fig 2). We use the radiometrically and geometrically corrected Sentinel-2 L1C Top-of-Atmosphere data product [34].

The public data portal includes site metadata queried from other publicly available datasets. The parameters include soil type information (clay and sand percentage, soil bulk density, and soil great group identity from OpenLandMap [36]), site elevation and slope (SRTM [37]), landform type (Global ALOS Landforms [38]), distance to nearest water bodies (OpenStreetMap [39]), and nearby population (WorldPop [40]).

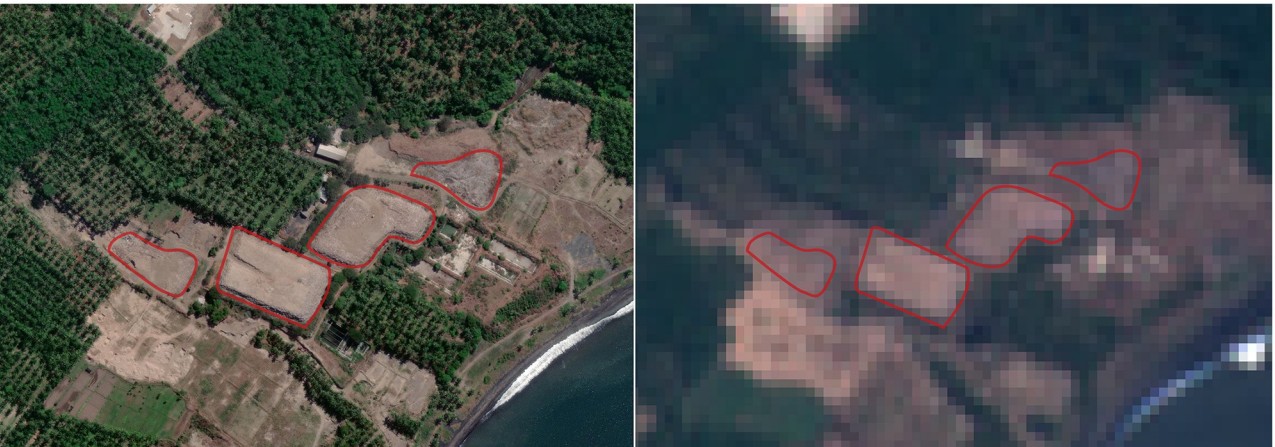

**Fig 2. Comparison of imagery used for validation and detection.** At left, high resolution imagery from Maxar Open Data Program [35] with the waste site boundary highlighted in red. At right, a red, green, and blue band composite of Sentinel-2 data. As seen here, site identification at 10-meter resolution is challenging for even human evaluators.

**Data labeling.** We began with a set of ten known waste sites in Bali, Indonesia, with hand-drawn boundaries. We select negative-class sites to capture the distribution of terrain in the target domain, while biasing toward features closer in spectra to waste than the dominant land cover type. In Bali, tropical forest dominates outside urban areas.

After trained models are run on a region, we add confirmed positive sites to the training set, and we evaluate dominant failure modes to select new negative-class sampling locations. In this way, we create a data sampling system that continually incorporates new information as the geographic scope increases through a bootstrapping process. After thirteen rounds of bootstrapping, the training dataset is made up of 213 locations containing waste and 345 regions without.

**Training data generation (Stage 1).** For each labeled site location, we extract all 2019 Sentinel-2 L1C (top-of-atmosphere) data in a $480 \times 480$ meter square patch around the site centroid, with Descartes Labs cloud and cloud-shadow masking (stage 1.1). The masks are broadly effective but leave behind residual cloud edges and haze. Haze in particular proved to be a consistent source of false-positive detections for early models. We experimented with various data compositing techniques. Because clouds and haze are bright, we were able to eliminate most wispy clouds and haze that escaped the cloud masks by taking a minimum instead of a median composite. In the final reckoning, the data input to the neural networks is Sentinel-2 L1C data with cloud and cloud-shadow masks, composited across a three-month window by selecting the minimum unmasked value for each pixel. To form a spectrogram, a three-month composite is paired with another composite at the same location, offset by six months. We then normalize the data per-spectral-channel across the training dataset. In total, the labeled patch dataset is composed of 1,770 positive samples and 3,104 negative samples.

The unlabeled dataset (stage 1.2) is constructed similarly but consists of randomly sampled patches from 10x10 km regions that are themselves selected for broad geographic diversity.

Pixel data (stage 1.3) is derived from a subset of the labeled patch dataset. Each pixel accompanies its temporal pair, and the two spectral profiles are concatenated into a single spectrogram of shape (2, 12). The pixel classifier training dataset contains 200,663 and 3,687,725 positive and negative class pixel spectrograms, respectively.

For the positive-class data, pixels are selected that fall within hand-drawn waste-site boundaries. This boundary is fixed, but the waste sites are dynamic. In some cases, sites may have

dormant periods where vegetation covers the waste. Assigning positive labels to these vegetated pixels in the training dataset would create overlap between the classes. As such, we use an NDVI threshold conservatively set at 0.4 to filter any pixels from the positive-class dataset that may be vegetated. This removes 6.6% of samples from the positive-class training dataset.

To generate a test dataset, positive-class data is sampled from within boundaries drawn around 50 known waste sites in Indonesia, between June, 2019, and June, 2021. Here too, the resulting data is likely contaminated with some vegetated and bare earth pixels, from times when formal waste site operators bury waste or shift active waste operations. We sampled negative test data from a range of land-cover classes in Indonesia, oversampling challenging modes such as cities and bare earth. The test dataset contains 18,473 and 312,557 positive and negative class pixel spectrograms, respectively. These are sampled from 259 positively-labeled patches and 274 negative-class patches.

## Model architectures and training (Stage 2)

**Pixel spectrogram classifier (Stage 2.1).** The pixel spectrogram classifier is a small convolutional neural network (CNN) with fully-connected layers following the convolutional block, as detailed in Fig 3.

The convolutional block generates features across band combinations at a single time point as well as differences in spectra across the points in the spectral time series. These features can then be synthesized in the fully-connected layers. The number of free parameters in the architecture are kept small in order to reduce the risk of overfitting to the relatively uniform training dataset.

Through parameter sweeps and model comparisons we set the default training to use the Adam optimizer with a learning rate of 0.001, a batch size of 128, and initialized layer weights using a Glorot uniform initializer. We did not observe a strong influence on model performance from training hyperparameters.

**Patch classifier (Stages 2.2–2.4).** To enrich the training the patch classifier is trained using a semi-supervised distillation process. An ensemble of classifiers is first trained on labeled data (stage 2.2). These classifiers make predictions on unlabeled data (stage 2.3) that are then combined and used as soft targets to train the final patch classifier (stage 2.4). The workflow for the distillation is drawn in Fig 4.

**Strong labelers (Stage 2.2).** An ensemble of 32 neural networks are trained on the labeled multispectral patch data. In contrast to the pixel classifier, the two temporal frames are concatenated along the spectral channel axis, so that on a patch 28 pixels square, the input tensor has shape (28, 28, 24). The models in the ensemble are trained with the same hyperparameters and on the same data, but the weights for each network are initialized with different random seeds in order to encourage model diversity.

The patch network is again a CNN, wider and deeper than the pixel classifier. The convolutional head contains three rounds of three convolutional layers followed by max pooling. These convolutional features are processed by a dense block. Architecture details are shown in Fig 5. During training, we augment input data with reflections and rotations and apply batch

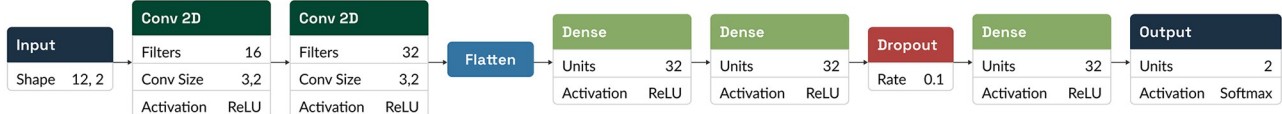

**Fig 3. The architecture of the pixel spectrogram classifier neural network.** Each block represents a layer or stage within the network. Dropout layers are only applied during training.

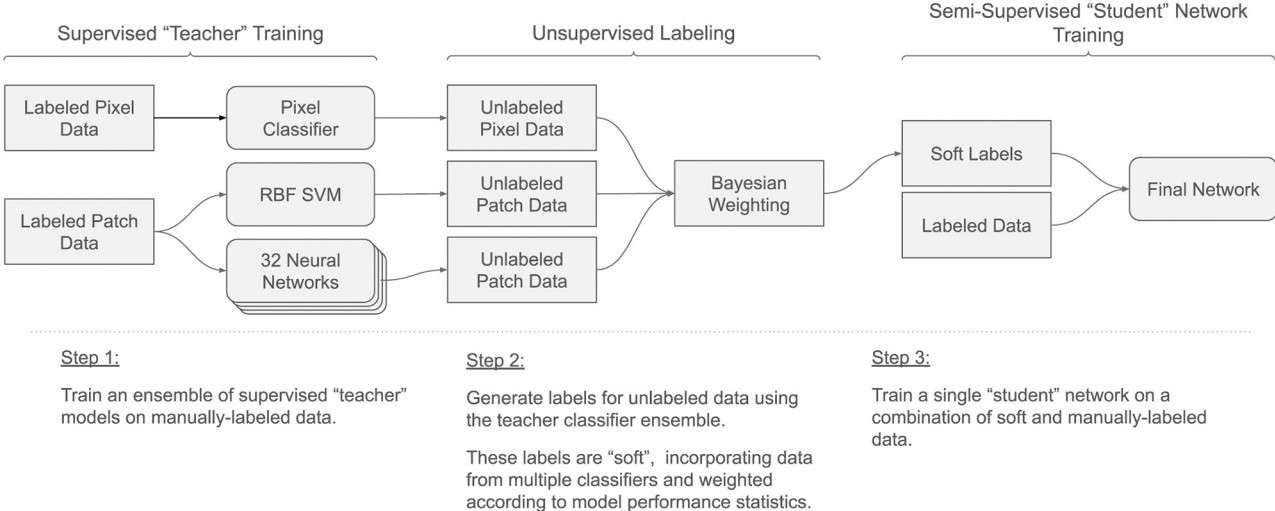

**Fig 4. The semi-supervised training process for the patch classification network.** Labels are generated using inputs from a variety of models, which are then combined into a single soft target that is used in combination with supervised data to train the final model.

normalization and dropout. During inference, these components are inactive. Aside from a scheduled learning rate, we train with the same hyperparameters noted for the pixel classifier.

**Soft labels (Stage 2.3).** Though support vector machines (SVM) are not often optimal for this form of image classification, we chose to incorporate predictions from an SVM to increase the diversity of outputs used to generate the soft labels. We flatten the labeled patch images from a shape of (28, 28, 12, 2) to a vector of length 18,816 and train a radial basis function kernel support vector machine to classify the data.

The neural network ensemble, the SVM, and the pixel classifier each generate predictions for all patches in the unlabeled dataset. Given that the patch classifier ensemble and the pixel classifier generate more than a single prediction, the outputs of these model types are processed into a single value through a series of heuristics.

The predictions of the neural network ensemble are combined by first converting them to binary values at a threshold of 0.5, and then selecting the mode of this set. This single value is multiplied by a metric of disagreement, formulated as $(1 - 2\sigma)$, where $\sigma$ is the standard deviation of the binary outputs. This allows a single label to represent richer information from the ensemble of networks.

The pixel classifier produces an individual prediction for each pixel in the patch. The patch is assigned a binary class if the mean value of all predictions within the patch surpass a threshold value of 0.02. This threshold was determined via an exhaustive sweep between 0.0 and 1.0, optimizing for average accuracy over the test set.

These individual model predictions are then unified into a single soft target through a Bayesian process. The neural ensemble serves as the first prior, which is then updated sequentially using predictions and training statistics from the SVM and the pixel classifier. The order in which our hypothesis is modified is arbitrary, since Bayesian updates are commutative. These soft targets are generated for every unlabeled patch and then used for training the student patch classifier. This technique is closely related to programmatic weak supervision labeling approaches [41].

**Student training (Stage 2.4).** A single student network is then trained on a combination of these soft-labeled data and hard targets from the human-labeled dataset. The network

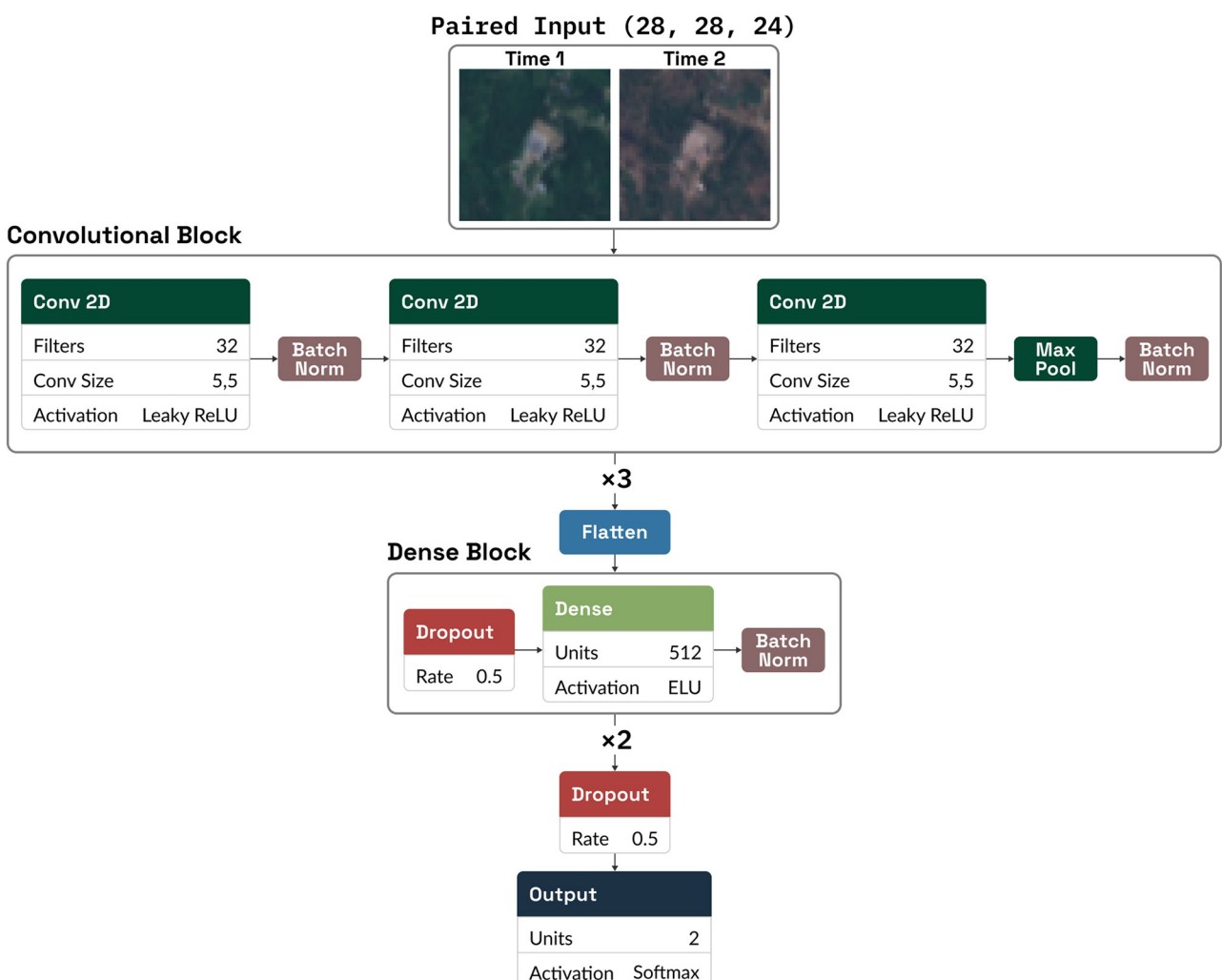

**Fig 5. The architecture of the patch classification neural networks.** The convolutional block is repeated three times, followed by two repetitions of the dense block. Dropout and batch normalization are only applied during training.

architecture and training strategy is the same as the ensemble of neural networks described previously (Fig 5).

### Waste site identification (Stage 3)

**Model inference (Stage 3.1).** The model is deployed on the Descartes Labs geospatial analytics platform. The Descartes Labs engine breaks a geographic region of interest into sub-tiles for parallel processing on a cluster of machines. The pixel classifier evaluates every pixel in the sub-tile, creating a heatmap of predicted waste locations. At the same time, the patch classifier is convolved across the scenes with a stride width of 8 pixels to accommodate for cases where a waste site might lie on the boundary between prediction windows. When deciding on a stride width, the tradeoff between the computational efficiency of a larger stride and resolution of a smaller stride must be weighed. An 8-pixel stride was deemed suitable with regards to computational efficiency, and little gain was found empirically on the test set from using a smaller stride. This generates a set of patch-based predictions for the presence of waste (Fig 6).

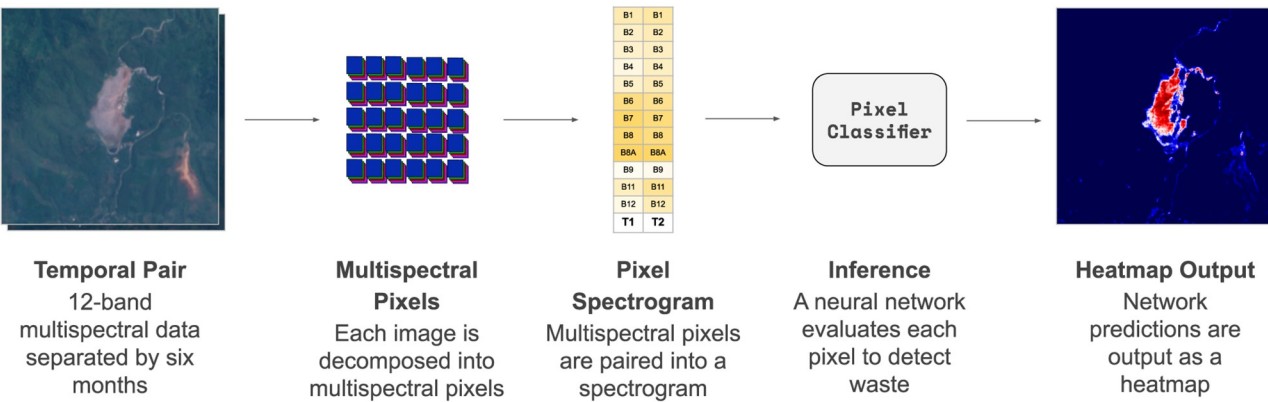

**Fig 6. The pixel classifier inference pipeline.** A temporally-paired set of Sentinel-2 data is broken apart into spectrograms to be classified by the pixel classifier neural network. This generates a heatmap of waste likelihood within the region, here visualized in red.

**Pixel classifier candidate detection (Stage 3.2).** To identify candidates from the pixel classifier heatmap, we mask any prediction below a threshold and detect connected clusters ("blobs") of pixels with high-valued predictions using the scikit-image Determinant-of-Hessian blob detection function (DoH) [42]. This eliminates single-pixel noise that may be present in the outputs and also produces a single coordinate for each candidate waste site.

The sensitivity of the candidate detection stage can be tuned by controlling the prediction threshold and the minimum sigma (min_sigma) value of the DoH algorithm. Tuning these parameters is a tradeoff between the precision and recall of detections. In absence of a database of known waste locations, we were not able to empirically sweep these values to find an optimum. Instead, we developed three sensitivity regimes detailed in Table 1 based on evaluations of candidates surfaced during model detection runs.

A higher classifier threshold leaves only regions which the networks assess as more certain to contain waste. The min_sigma parameter is used by the DoH blob detection and controls the minimum standard deviation of the Gaussian blur kernel in the algorithm. As such, a lower min_sigma parameter enables smaller and fainter blobs to be returned as candidates.

To put these values in context, a $3 \times 3$ pixel region with predicted values of 0.7 would be detected as a candidate using a min_sigma value of 3.5, but would not be identified using a min_sigma value of 5.0. Similarly, a $2 \times 3$ pixel region with predictions of 0.7 would not be surfaced at either min_sigma value. If this $2 \times 3$ region had predicted values of 0.8, it would be identified using a min_sigma of 3.5. The interplay between classifier thresholds and min_sigma values are shown in Fig 7.

**Patch classifier candidate validation (Stage 3.3).** Because the pixel classifier has no ability to incorporate spatial information, it is liable to misclassify objects that share a spectral profile with waste. We have seen that the pixel classifier may positively identify the plastic roofs on greenhouses given their similar spectral profile to plastic waste in dump sites. For this reason, each pixel classifier candidate is checked by the patch classifier predictions for that location. If

**Table 1. Parameters for different candidate site generation sensitivity modes.**

|  | Pixel Threshold | Min Sigma | Patch Threshold |
|---|---|---|---|
| Low | 0.9 | 5.0 | 0.6 |
| Med | 0.6 | 5.0 | 0.6 |
| High | 0.6 | 3.5 | 0.3 |

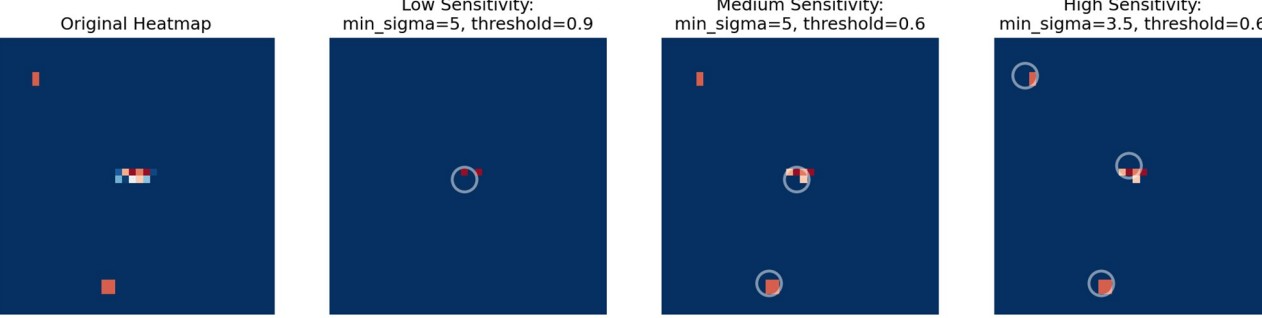

**Fig 7. Sensitivity modes of the blob detection stage.** A pixel classifier heatmap is shown with redder pixels indicating a higher classification score. Candidates identified following blob detection are circled. Candidate detection sensitivity modes are shown in each panel.

any of the patch classifier predictions is greater than a threshold, the candidate site is accepted. Again, sensitivity can be tuned by adjusting this patch classifier threshold (Table 1).

**Manual site verification (Stage 3.4).** Each waste site identified by the detection system is evaluated by a trained evaluator. Details on this process are described in the output validation section. Candidates that pass this evaluation process are considered confirmed.

**Metadata generation (Stage 3.5).** We generate additional metadata for each confirmed waste site. Geophysical and population data is generated by querying external datasets, listed in the data sources, at the location or region surrounding the waste aggregation. To compute each site's distance to the nearest waterway, we use the OpenStreetMap API to return all waterways within a 5 km radius. For each waterway found, we compute the nearest distance between the waste site centroid and the geometry of each waterway. The minimum value of this set is returned as the site's nearest distance to a waterway. If no waterways are found within a 5 km radius, this location is assigned a value of >5 km.

## Site footprint monitoring (Stage 4)

For each confirmed waste aggregation we compute the footprint of the site and how it changes through time. To do so, we extract composited mosaic pairs as previously described (Stage 1.1). The composites are extracted every month for the full extent of the Sentinel-2 catalog, reaching from mid-2017 through January, 2021. A pixel-classifier prediction is computed for every pixel within the patch, which produces a set of heat map predictions of waste locations for each time point in the dataset.

Generating site boundaries is an unforgiving task. Misclassification of a single 10-meter Sentinel-2 pixel may represent a substantial fluctuation in the total site area, and classification at single time points is prone to noisier predictions than we achieve after time averaging in the detection stage. Thus, we generate and apply a rolling prediction mask to minimize the influence of outliers that are more often present when evaluating at a single time point. This mask is computed as a thresholded rolling median of the eight following predictions and is applied to the current frame. Applying a mask utilizes the information present in the time series prediction in order to generate a region of interest and filters outliers, while still allowing the outputs at the current time step to establish the current waste location. The length of the mask window is adjustable, with a mask length of eight selected to balance between responsiveness to change and filtering strength.

This masked prediction frame is thresholded, and contours surrounding the binary output are generated [43]. These contour boundaries establish the waste site footprint at monthly intervals. Because the pixel classifier takes temporal inputs that are offset by a six month

period, contours tend to represent the locations of waste that are present at both time points. This process is repeated for each image in the dataset to create a record of site footprints through time.

## Output validation

Points that are positively classified by both networks are denoted as "candidate sites." To confirm the accuracy of the identifications, these candidates are evaluated by a curator using publicly-available data. Predominantly, validation is done through analysis of very-high resolution satellite data hosted on the Google Earth and Mapbox platforms. This imagery is a composite collected from multiple sensing platforms, but Very-High-Resolution imagery (ranging from under 60 cm and up to 5 m in resolution) is available in most cases [44]. At this resolution, one can identify characteristic traits that are frequently seen in waste sites. These may include a high frequency texture, pale gray or brown coloration, roads and paths leading to active dumping areas, machinery for moving or dumping waste, or plumes from burning.

Depending on availability, sites may also be verified using other data sources. In Google Street View imagery, reviewers can clearly see the individual waste items as well as the aggregation as a whole. Of the identified sites in Indonesia, 126 sites have nearby Street View imagery. In other cases, data from Planet Planetscope and OpenStreetMap can be used as additional sources of information for site confirmation.

This process has the benefit of limiting the number of false positives. However, it is imperfect. The high-resolution data is frequently older than the data used for detection. Thus, a candidate waste site might be rejected if it is newly created. Additionally, waste aggregations are not always obvious to identify even in the high resolution images. If there is any uncertainty regarding whether a candidate is a waste site, the site is rejected.

## Results

### Waste site detection and system performance

**Indonesia.** We evaluated every $10 \times 10$ Sentinel-2 pixel captured in Indonesia ($1.81 \times 10^{6}$ $km^2$) at nine time steps between January, 2019 and March, 2021. This produced 163 billion predictions at the pixel level and 623 million classifications of patches. To reduce variance, we average the time-step outputs to arrive at a final assessment of the presence of waste.

In total, the model detected 374 waste aggregation sites across Indonesia (Fig 8) that trained reviewers were able to confirm through a manual review process that is detailed in the output validation section. This is more than double the number cataloged waste sites in known databases. The nature of detected sites vary, though identifications are predominantly formal government-run open waste sites and small-scale informal dumpsites.

Using data from the Indonesian Ministry of Public Works and Public Housing [46] and OpenStreetMap, we compiled a list of 184 waste sites operating across Indonesia. Though a complete set of waste locations is not known, we use this subset of sites to evaluate the false negative rate for the model. The system had a recall rate of 80% in a high sensitivity configuration, and a 40% recall rate in low and medium sensitivity modes (Table 2). The system detects about three previously unknown waste sites for every site it misses. As another point of reference, we see that the number of waste sites detected correlates well with the population within the region of evaluation (Pearson correlation, 0.991).

**Southeast Asia.** We also ran the system across all countries in Southeast Asia. Because the model received no tuning or additional training data to expand beyond Indonesia, we ran the pipeline in a low sensitivity configuration. We detected and confirmed a total of 996 waste aggregation sites in Southeast Asia (Fig 9). This is a nearly three-fold increase over the number

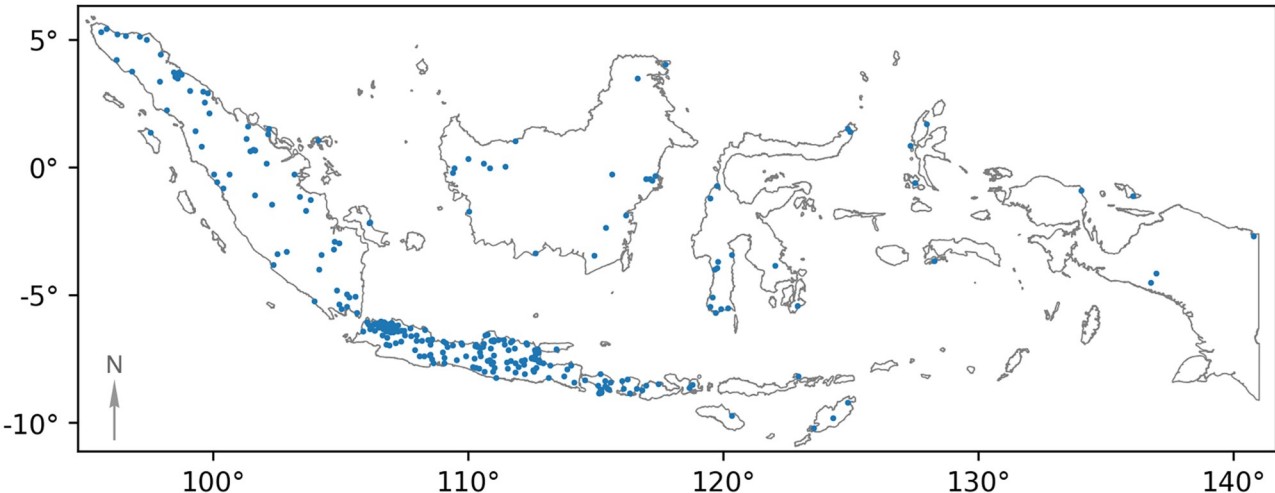

**Fig 8. Locations of confirmed waste sites identified in Indonesia.** Map created with GeoPandas [45] with country boundary provided by the Database of Global Administrative Areas.

of recorded waste sites listed in these countries on OpenStreetMap. 53% of candidate locations produced by the pixel classifier and confirmed by the patch classifier were validated by human evaluators as waste aggregations (Table 3).

Once again, the number of detected waste sites within a country correlates well with its population (Pearson correlation, 0.960). Though this relationship is influenced by the country's waste management practices, it provides grounding that the results are sensible in the absence of ground truth information on waste site distribution.

## Site-specific metrics

**Site proximity to waterways.** We find that the centers of 19% of waste sites in Southeast Asia are located within 200 meters of a waterway or waterbody listed on OpenStreetMap, and more than half are within 750 m (Fig 10). For sites that are located within 5 km of a waterbody, the median distance is 706 m.

**Table 2. Statistics on waste site detection in Indonesia separated by island and sensitivity mode.** Population data from Indonesian Central Bureau of Statistics [47].

| Island | Population (Millions) | Sensitivity | Known | Known Sites Detected | Recall | Newly Detected | Detected / Known | Total Detected | |
|---|---|---|---|---|---|---|---|---|---|
| Java | 152 | high | 50 | 39 | 78% | 193 | 464% | 232 | |
| Bali | 4.32 | high | 6 | 6 | 100% | 12 | 300% | 18 | |
| Sumatra | 58.6 | medium | 45 | 21 | 47% | 45 | 147% | 66 | |
| Sulawesi | 19.9 | low | 34 | 13 | 38% | 2 | 44% | 15 | |
| Kalimantan | 16.6 | low | 28 | 7 | 25% | 12 | 68% | 19 | |
| Papua | 5.44 | low | 5 | 1 | 20% | 4 | 100% | 5 | |
| East Nusa Tenggara | 5.33 | low | 4 | 2 | 50% | 3 | 125% | 5 | |
| West Nusa Tenggara | 5.32 | low | 3 | 3 | 100% | 2 | 167% | 5 | |
| Lombok | 3.76 | low | 2 | 2 | 100% | 3 | 250% | 5 | |
| Maluku | 1.85 | low | 7 | 3 | 43% | 1 | 57% | 4 | |
| **Total** | 273 | - | 184 | 97 | - | 277 | - | 374 | |
| **Mean** | 27.3 | - | - | - | 53% | - | 203% | - | |

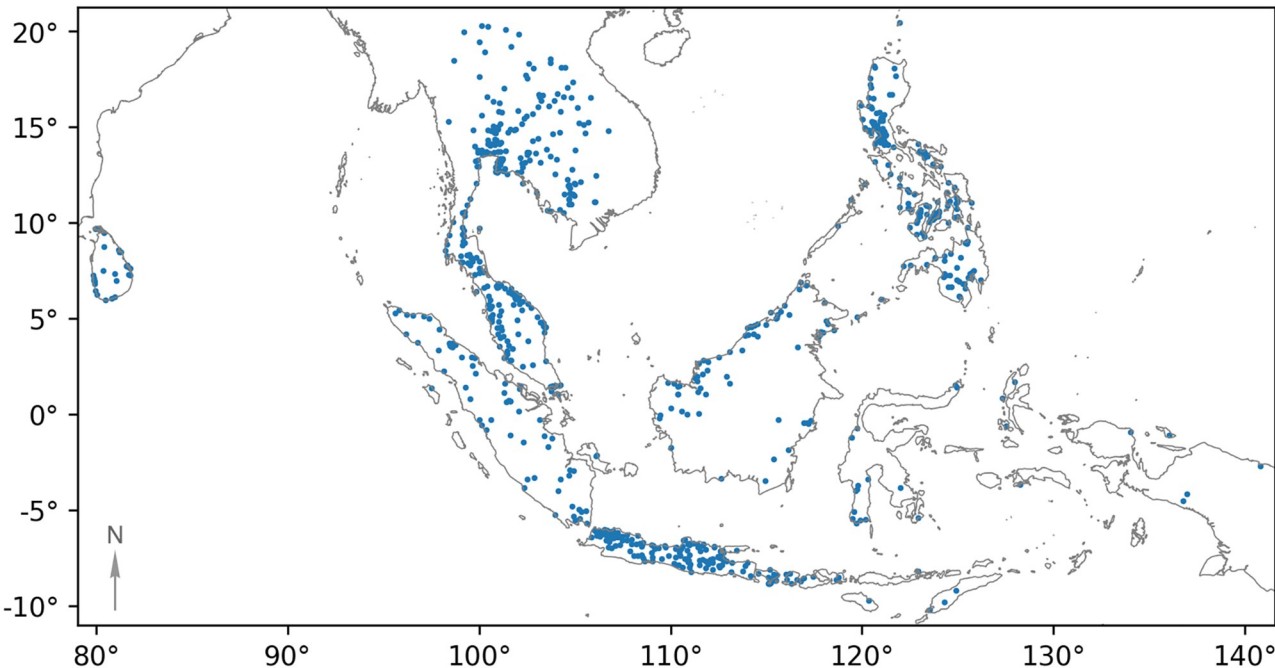

**Fig 9. Locations of confirmed waste site detections across Southeast Asia.** The map was created with GeoPandas [45] using country boundaries sourced from Natural Earth.

We also identify a number of waste sites situated directly on the banks of rivers. Referencing high-resolution satellite imagery, it is clear to see the waste overflowing retaining structures and spilling directly into the waterways. Sites can be seen located on the coast, with waste collapsing into rivers, or with waste mounds eroded by river flow. Though these cases are

**Table 3. Count of sites detected by country in Southeast Asia and comparison with known sites listed on OpenStreetMap.** Population data from United Nations' World Population Prospects, 2020.

| Country | Population (Millions) | Confirmed Detections | Candidates | True Positive Rate | Listed Sites (OSM) | Detected / Known | |
|---|---|---|---|---|---|---|---|
| Vietnam | 96.6 | 96 | 133 | 72% | 14 | 686% | |
| Thailand | 71.5 | 154 | 228 | 68% | 29 | 531% | |
| Myanmar | 53.4 | 50 | 131 | 38% | 13 | 385% | |
| Malaysia | 33.2 | 101 | 232 | 44% | 39 | 259% | |
| Sri Lanka | 21.7 | 28 | 15 | 65% | 14 | 200% | |
| Cambodia | 16.4 | 48 | 130 | 37% | 2 | 2400% | |
| Laos | 7.32 | 17 | 53 | 32% | 5 | 340% | |
| Timor Leste | 1.30 | 1 | 4 | 25% | 0 | - | |
| Brunei | 0.442 | 4 | 10 | 40% | 1 | 400% | |
| Indonesia | 272 | 374 | - | - | 117 | 320% | |
| Philippines | 112 | 118 | - | - | 116 | 102% | |
| Singapore | 5.91 | 5 | - | - | 3 | 167% | |
| **Total** | 692 | 996 | - | - | 353 | - | |
| **Mean** | 57.6 | - | - | 53% | - | 282% | |

Indonesia, the Philippines, and Singapore do not have the number of candidates as these analyses were conducted prior to compiling this information in the validation pipeline.

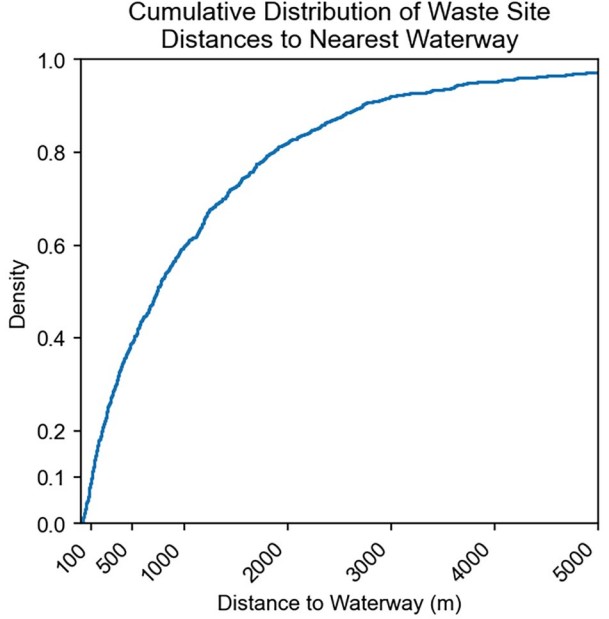

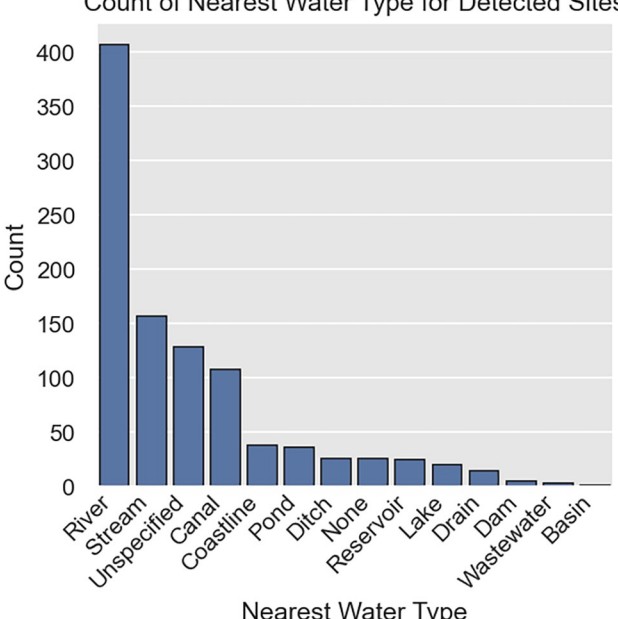

**Fig 10. Waste sites are frequently found near waterways.** Cumulative distribution showing the proportion of waste sites as a function distance to the nearest waterway or water body at left. Nearest water type, as listed on OpenStreetMap, is shown at right.

numerous, sites of note are listed in Table 4. Some sites seen leaking into water later show signs of remediation, in the form of retaining walls, rebuilt banks, or waste burial, suggesting that they are in fact environmental hazards and recognized as such by local authorities.

**Footprint monitoring.** Using the pixel classifier, we generate monthly site boundaries. It is difficult to quantitatively assess their accuracy and precision in the absence of ground truth data. Visual inspection of imagery does not suffice, because human labelers cannot reliably delineate boundaries between bare earth and waste in high resolution imagery. Qualitatively, waste site boundaries frequently visually match historical imagery (Fig 11). They also exhibit the misclassification modes of the pixel classifier.

In terms of the mean footprint area across time, 38% of detected waste sites are smaller than 0.1 ha, equivalent to a square area about 30 meters on a side, and 82% of sites are smaller than 0.5 ha (Fig 12). The mean area of a single site in the region is 0.47 ha (SE 0.044). As a reference for dumping ground size, the footprints of known formal waste sites in Bali range from 0.51 to 4.5 ha. Given that managed waste sites empirically tend to be larger, the number of

**Table 4. A selection of notable sites with visable overflow of waste into waterways.**

| Site | Latitude | Longitude | Site URL |
| --- | --- | --- | --- |
| Philippines | 16.089 N | 120.356 E | https://globalplasticwatch.org/map?site=8f6945231090416 |
| Vietnam | 21.118 N | 106.419 E | https://globalplasticwatch.org/map?site=8f8c16aaccc6540 |
| Indonesia | 6.141 S | 106.616 E | https://globalplasticwatch.org/map?site=8f8c106c240b149 |
| Indonesia | 6.206 S | 107.034 E | https://globalplasticwatch.org/map?site=8f8c104c4052382 |
| Indonesia | 6.591 S | 107.741 E | https://globalplasticwatch.org/map?site=8f8c16aaccc6540 |
| Sri Lanka | 7.771 N | 81.601 E | https://globalplasticwatch.org/map?site=8f6111d92a4c286 |

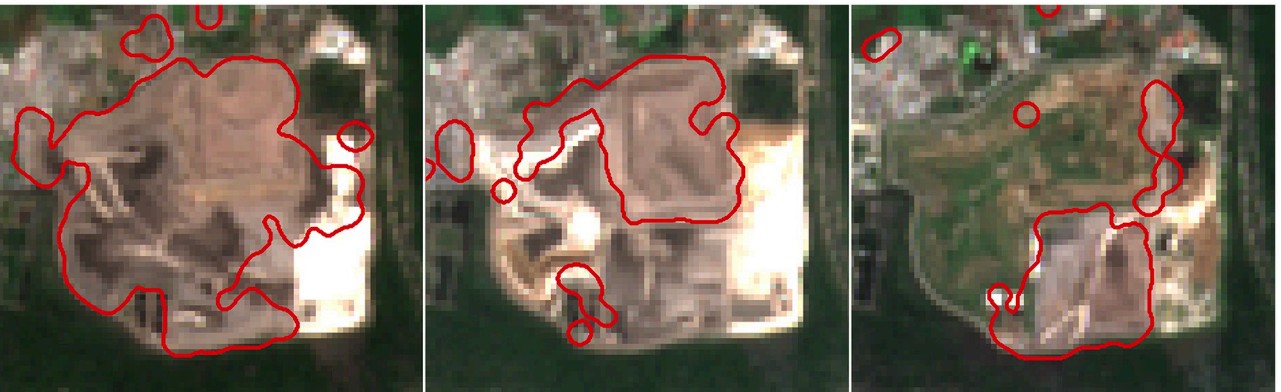

**Fig 11. Example of auto-generated waste site footprint.** Site boundaries of a waste site in Bali, Indonesia (8.722 S, 115.221 E) visually correspond to its development and subsequent reduction. Sequence runs May, 2018; August, 2018; and May, 2020. The imagery in this figure comes from Sentinel-2, though these changes can also be seen and validated using high-resolution imagery.

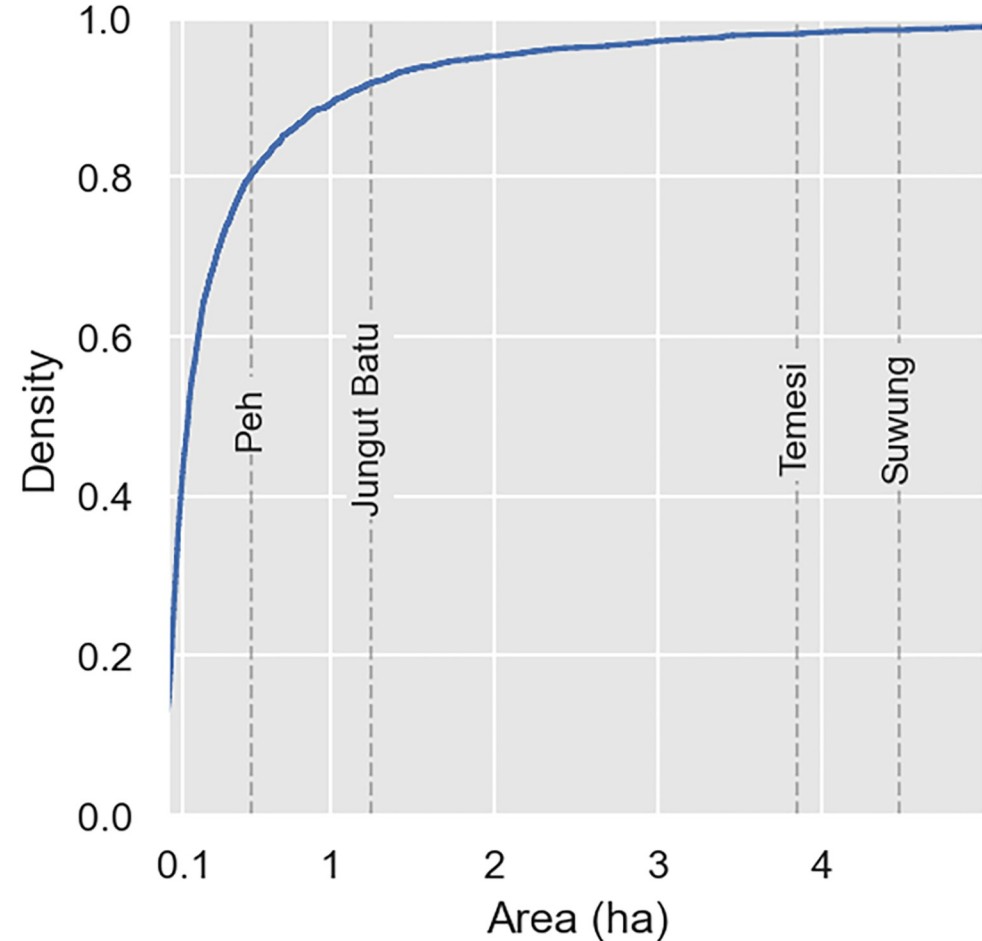

**Fig 12. Cumulative distribution of average site area for detected waste sites in Southeast Asia.** As a reference of waste site area, the areas of four formal waste sites in Bali are plotted (left), and shown (right).

identified sites smaller than 0.5 ha may indicate that the majority of detections in this work are informal dumping grounds.

## Individual model accuracy

In evaluating on a fully-withheld test dataset, the component single pixel and patch-based neural networks prove to be highly performant, with f1 scores over 90% (Table 5). Performance persists across individual land-cover classes (forest, farm, etc.) within the negative-class test data, with city and bare earth emerging as relatively challenging cover types. Details on the construction of the test dataset used in these evaluations are given in the methods.

To identify the importance of input data features, we compared the performance of the 12-band spectrogram pixel classifier against models with selectively reduced inputs (Table 5).

Including temporal information through a spectrogram input improved the true positive rate from 66.64% to 71.99%, and increased the true negative rate from 99.65% to 99.97%. Despite hiding in the decimal-percentage places, this increase in true negative rate represents an order of magnitude improvement in false negative suppression. The domain of operation for the system is highly class imbalanced, with approximately ten million true negative pixels for each single true positive pixel. These seemingly fine numerical margins of improvement make the difference between a practically useful system and one where true positives are lost in the noise of false detections.

Broad spectral coverage is also seen to be essential for waste identification. Reduced-spectrum networks that took only RGB or RGB+NIR bands as input showed only a minimal capacity to identify waste, as evidenced by true positive rates under 30%.

Finally, the neural network demonstrated a greater capacity for classification compared to a random forest trained on the per-pixel data, with an unweighted f1 score of 90.46% vs. 77.39%.

Though the patch classifier functions only to cross-validate the pixel classifier candidates, it too has a high level of classification accuracy. We evaluated the performance of this semi-supervised single network versus its teacher ensemble of 32 supervised networks, and found that the student network had an f1 score of 97.56% vs. the ensemble's 95.31%. Of course, the single network also offers the benefit of more efficient inference as compared to the ensemble (Table 5).

**Table 5. Performance metrics for pixel- and patch-based classifiers.**

| | | | Accuracy | | | | | | | | |
|---|---|---|---|---|---|---|---|---|---|---|---|
| | | | | | | Land Cover Type | | | | | |
| | | | Positive | Negative | f1 | Forest | Farm | River | City | Bare Earth | Beach |
| Pixel | **Spectrogram** | | **71.99%** | **99.97%** | **90.46%** | **100.0%** | **100.0%** | **100.0%** | **99.87%** | **99.97%** | **100.0%** |
| | Atemporal | | 66.64% | 99.65% | 88.05% | 100.0% | 99.87% | 99.74% | 99.30% | 99.38% | 99.89% |
| | RGB Only | | 17.27% | 99.01% | 61.45% | 100.0% | 100.0% | 100.0% | 97.43% | 97.90% | 100.0% |
| | RGB-IR Only | | 27.93% | 99.69% | 69.85% | 100.0% | 99.89% | 100.0% | 99.11% | 99.55% | 100.0% |
| | Random Forest | | 44.49% | 99.28% | 77.39% | 100.0% | 100.0% | 99.66% | 97.54% | 99.31% | 99.99% |
| Patch | **Student** | | **95.62%** | **99.61%** | **97.56%** | **100.0%** | **100.0%** | **100.0%** | **98.28%** | **100.0%** | **100.0%** |
| | Teacher Ensemble | | 90.88% | 100.0% | 95.31% | 100.0% | 100.0% | 100.0% | 100.0% | 100.0% | 100.0% |

The two neural networks deployed in our waste site detection pipeline are indicated in bold. Above the line are per-pixel classifiers. Below the line are patch-based classifiers. Description of the comparison networks is given in the results and methods sections.

## Discussion

This research establishes the first comprehensive dataset on the distribution and characteristics of plastic aggregation and waste sites within Southeast Asia by remote sensing. It also demonstrates an architecture for using neural network-based systems to consistently and extensibly detect and monitor plastic and waste aggregations in Sentinel-2 data. In Indonesia, we more than doubled the count of known sites and found 277 waste sites that do not exist in public databases (Table 2). We then expanded the model to identify and validate nearly a thousand plastic and waste aggregation sites across Southeast Asia (Table 3). The results of this work are presented in an open data portal at https://globalplasticwatch.org/, providing information to the public, governments, non-government organizations, and multiple industries about the spatial and temporal characteristics of waste aggregation sites. This data can be used to inform upstream interventions and to prevent further plastic pollution, as well as inform other mitigation efforts, waste management strategies, cleanup campaigns, and monitoring efforts. This open data platform offers both assessment and monitoring of terrestrial pollution at a scale that has not been realized previously.

As discussed in the methodology, the validation of this data product relies on high-resolution satellite or aerial imagery. As such, detections present in the curated dataset are bounded by the quality and recency of the validation data source. In practice, this means that detected waste sites that appear after high-resolution validation data was collected will be rejected in the human validation stage. Additionally, validators may also improperly reject detected waste site candidates that are small, visually indistinct, or fall within areas where publicly-accessible validation data is of low quality. If desired, the Sentinel-2 detection system described in this work could inform a "tip and cue" system. Here, waste sites identified by the models could be added to a set of targets for high-resolution data collection or in-situ validation.

This work may serve as a template for utilizing deep learning for environmental monitoring and detection systems. In particular, we demonstrate model and system architectures for incorporating multiple dimensions of remotely sensed information. Convolving learnable filters across spatial, spectral, and temporal dimensions combines these signals in ways that would be nearly impossible to envision for handcrafted algorithms and greatly boosts system performance (Table 5). The neural networks also show a capacity for geographic robustness, functioning in unseen countries across the geographically varied Southeast Asian region without additional tuning (Table 3).

Labeled data for environmental monitoring is often scarce, and this work demonstrates strategies for training neural networks in data-poor conditions. The work began with only 10 known waste locations, which would typically be considered an insufficient dataset to train heavily-parameterized models like neural networks. We amplified the amount of training data by sampling at multiple time points and continuously adding to the dataset as new detections were validated and new failure modes were identified. This bootstrapped sampling enhanced the quantity and diversity of training data and continually improved the training set through time. The bootstrapped training dataset is self-reinforcing as the geographic scope expands and new waste locations are identified.

Beginning with a per-pixel classifier architecture facilitated early progress. The classifier is forced to learn spectral and temporal patterns, minimizing bias and overfitting towards waste site structure that would likely have been seen in a spatial classifier. The second-stage temporal patch classifier is able to rule out candidates that spectrally match waste site profiles but are structurally different from waste sites. For example, the single pixel classifier initially identified plastic greenhouses as a plastic aggregation site, but second-stage spatial classification was able to distinguish the difference such that only plastic aggregation sites were identified. Finally, the

use of a semi-supervised noisy student distillation process improved the quality of the patch classifier. While distillation is often used to shrink the size of a model, our work showed that it also improved model performance and robustness (Table 5). Using this technique in cases with limited training data may be useful for application in other earth observation tasks.

Though we have been able to develop models of impressive capability given limited labeled data, the detection system is not guaranteed to perform outside of the distribution of the data used to train the models. Since the training data comes from only Southeast Asia, we anticipate that more training data will be needed to identify waste sites in regions with ecosystems that are dissimilar to those in the domain of the current training data.

In future work, we will run this system globally. In order to accommodate this geographic expansion, we will continue to use the same data engine collection procedure for the new regions. As the domains of evaluation grow, the model sizes may need to increase as well in order to capture the additional training data variance.

This work has implications for the integration of science into decision-making for plastic pollution and waste management. For example, the data illustrates that waste aggregations are often nearby or adjacent to waterways. More than half of sites are within 750 m, and 19% are within 200 m of a waterway (Fig 10). This highlights the role of these areas as a potential link between terrestrial waste aggregations and aquatic plastic pollution. Communities are burdened with waste management. After disposal, waste loses traceability and transparency. By identifying aggregation points, this observation system might help communities better understand pollution pathways. Researchers may also be able to use this data as a complement to other data being collected (e.g., litter data), and/or to validate or improve waste generation and management models, thereby improving estimates.

This data may also be used to prioritize remediation of high-risk waste sites. The data illustrates where waste aggregation is already occurring. In many of these cases in South and Southeast Asia, an informal waste management system already exists. Instead of closure, these areas could be targeted for inclusive infrastructure development since there is existing informal collection, aggregation, and management occurring. Informal workers are knowledge-holders that would contribute to both the development of, and participation in, a waste management system that is more protective of their health and the environment. Because the data allows for monthly monitoring of waste site presence and boundaries (Fig 11), the effectiveness of management interventions can be measured and monitored.

With all data open and available, non-governmental organizations, community leaders, and members of the public will be able to use it to advocate for changes to policies and practices in their communities. However, engagement of local government and key stakeholders is absolutely critical to the use of the data for context-sensitive interventions. The intention of this work is to expand it to the global scale, as plastic pollution knows no boundaries, and we are working to both refine detection of smaller waste aggregations and improve recall in new geographies. Although this work is groundbreaking from an open access assessment and monitoring scale, an Earth observation system is only one piece of an integrated approach to addressing plastic pollution. Partnering this data with a more holistic approach, including upstream interventions, is essential to effectively serve communities and reduce plastic entering our oceans.

## Supporting information

**S1 Fig. Detailed methodological pipeline.** Diagram showing the methodology pipeline in greater detail. Elements are colored according to type. Processing stages are shown in gray, processing configuration parameters in red, and outputs in blue. Major pipeline components

grouped and contained within dashed outlines.
(TIF)

## Acknowledgments

This work would not have been possible without the support of the full team at Earthrise Media. Particular thanks to Glynis Lough for structuring thought and action, Daniel Israel for deploying the system and validating candidates, Stephen Downs for creative direction of the data exploration platform, and Tom Ingold and Tom MacWright for building the website infrastructure.

## Author Contributions

**Conceptualization:** Caleb Kruse, Dan Hammer, Fabien Laurier.

**Data curation:** Caleb Kruse, Edward Boyda.

**Formal analysis:** Edward Boyda.

**Funding acquisition:** Dan Hammer, Fabien Laurier.

**Investigation:** Caleb Kruse, Edward Boyda, Sully Chen, Krishna Karra.

**Methodology:** Caleb Kruse, Edward Boyda, Sully Chen, Krishna Karra.

**Project administration:** Dan Hammer, Fabien Laurier.

**Resources:** Dan Hammer, Fabien Laurier.

**Software:** Caleb Kruse, Edward Boyda, Sully Chen, Krishna Karra, Tristan Bou-Nahra.

**Supervision:** Dan Hammer, Jenna Jambeck, Fabien Laurier.

**Validation:** Caleb Kruse, Edward Boyda, Dan Hammer, Jennifer Mathis, Taylor Maddalene, Jenna Jambeck, Fabien Laurier.

**Visualization:** Caleb Kruse.

**Writing – original draft:** Caleb Kruse, Edward Boyda, Sully Chen, Jennifer Mathis, Taylor Maddalene, Fabien Laurier.

**Writing – review & editing:** Caleb Kruse, Edward Boyda, Sully Chen, Jenna Jambeck, Fabien Laurier.

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
