## [Decision Letter · Decision Letter 0]

19 Sep 2022

PONE-D-22-22011Satellite monitoring of terrestrial plastic wastePLOS ONE

Dear Dr. Laurier,

Thank you for submitting your manuscript to PLOS ONE. After careful consideration, we feel that it has merit but does not fully meet PLOS ONE’s publication criteria as it currently stands. Therefore, we invite you to submit a revised version of the manuscript that addresses the points raised during the review process.

I would ask you make the edits requested by the reviewers, with careful attention given to the formatting notes (both reviewers agreed your non-standard ordering and integration of sections was confusing), an improvement to the discussion (especially with regard to including future directions and limitations), and - equally important - ensuring your github repository is open so we can retrieve the data underlying your findings.  The reviewers and I were unable to access that repository, and I am unable to recommend publication until it is made accessible. In your response, please provide a point-by-point rebuttal to each reviewer, alongside specific edits you made to the piece in response to each comment. Broadly, I enjoyed this piece, and found the application of neural networks to this topic to be quite novel.  I look forward to your resubmission.

We look forward to receiving your revised manuscript.

Kind regards,

Daniel Runfola

Academic Editor

PLOS ONE

Journal Requirements:

"This work was supported and funded by the The Minderoo Foundation."

"Earthrise Media and JJ received funding from the Minderoo Foundation (https://minderoo.org/) for this work. The funders had no role in study design, data collection and analysis, decision to publish, or preparation of the manuscript."

5. We note that Figures 1,2,4,5,6,8,11 and 12 in your submission contain [map/satellite] images which may be copyrighted. All PLOS content is published under the Creative Commons Attribution License (CC BY 4.0), which means that the manuscript, images, and Supporting Information files will be freely available online, and any third party is permitted to access, download, copy, distribute, and use these materials in any way, even commercially, with proper attribution. For these reasons, we cannot publish previously copyrighted maps or satellite images created using proprietary data, such as Google software (Google Maps, Street View, and Earth). For more information, see our copyright guidelines: http://journals.plos.org/plosone/s/licenses-and-copyright.

a. You may seek permission from the original copyright holder of Figures 1,2,4,5,6,8,11 and 12 to publish the content specifically under the CC BY 4.0 license.  

Natural Earth (public domain): http://www.naturalearthdata.com/\\

Reviewers' comments:

Reviewer's Responses to Questions

**Comments to the Author**

1. Is the manuscript technically sound, and do the data support the conclusions?

Reviewer #1: Yes

Reviewer #2: Yes

2. Has the statistical analysis been performed appropriately and rigorously? 

Reviewer #1: Yes

Reviewer #2: Yes

3. Have the authors made all data underlying the findings in their manuscript fully available?

Reviewer #1: Yes

Reviewer #2: No

4. Is the manuscript presented in an intelligible fashion and written in standard English?

Reviewer #1: Yes

Reviewer #2: Yes

5. Review Comments to the Author

Reviewer #1: This paper discusses the uses of neural networks to analyze spectral, spatial, and temporal components of Sentinel-2 satellite data for identifying terrestrial clusters of waste. The topic is very relevant, and I read the paper with great interest as I have been hoping that someone would do such a study sooner than later.

I found the paper to be well written with further scope for improvement. Methodologically the manuscript is novel and one of the few that uses neural networks for identifying aggregated waste. This topic would greatly interest city planners, urban managers, aid agencies, donors, and in-country policymakers. Therefore, I would recommend the paper be published with the suggested changes.

1. Introduction section included methods therefore, it might be useful to present a flowchart showing the steps involved.

2. Line 31, The manuscript includes a comprehensive literature review. I would advise referring to similar work funded by the JMSS featured here: https://www.undp.org/vietnam/blog/remote-sensing-smart-city-solution-municipal-waste-management

3. Line 77: The interchangeable use of plastic and waste seems confusing. How do we know it is only plastics? The landfill sites consist of mixed waste, in most of developing countries, waste segregation is still a challenge. It might be useful to highlight the heterogeneity in the waste composition and discuss the percentage of plastics present across the geographical regions.

4. Line 175: What do you mean by “South Asian biome.” I am not aware of any such biome. South Asia is a geographical construct.

5. Line 453. Google and Mapbox host the base maps. Please provide the data description and resolution.

6. The discussion section is very weak. It should highlight future work and ways the results could be further improved.

7. The paper doesn’t seem to follow the PLOS ONE format. Please refer to the journal outline and organize the sections accordingly. This will help improve the readability of the paperhttps://journals.plos.org/plosone/s/submission-guidelines

Reviewer #2: Overall I found this to be a well produced article that combines meaningful work with thoroughly described methods that will assist reproduction and future efforts. Most of my comments are fairly minor.

Abstract - Given the focal area of South East Asia, prominently describing this as working at continental scales seems slightly out of place. Maybe include that this can be expanded to continental scales towards the end of the abstract instead?

Line 3 - While you do go into more detail about the impacts of plastic waste, a good cite early on for “via pathways that are not fully understood” would help build a foundation of where this work fits into both the broader literature and real world applications.

Line 69 - It seems like the paper was misformatted as the Results (and Discussion) sections come before the Methods (and data).

Table 1 - The amount of new detections and related rates are heavily influenced by Java. While this is just a piece of the results, I think it would be good to have some discussion of whether this is driven by island size, sample selection, location of actual waste, etc.

Line 285 - What is the actual size (km) of Sentinel 2 patches around the centroids?

Line 303 - How many iterations of the data engine did this go through? While this is iterative / ongoing for production, I assume there was a “final” number of iterations used for the results in this paper.

Line 378 - A cite for the Bayesian updates would be helpful.

Table 4 - Can you add more details on how you came up with the settings for the sensitivity modes?

The GitHub repository does not seem to be public at the moment. Was this kept private because the paper is under review?

6. PLOS authors have the option to publish the peer review history of their article (what does this mean?). If published, this will include your full peer review and any attached files.

Reviewer #1: No

Reviewer #2: No

---

## [Author Response · Author response to Decision Letter 0]

11 Nov 2022

This information has been provided in the "Response to Reviewers" document and contains more readable formatting. I am copying the contents of this document below:

Responses to PLOS ONE First Review Comments

11/2/2022

Response From the Authors

Thank you to the editor and reviewers for evaluating this paper. It’s clear that there is an understanding of this work and the intentions of the paper. The comments and subsequent changes have made the paper stronger. 

In this document, we have provided responses to each of the comments. If we have made a change to the paper, the location of the change in the re-submitted version of the manuscript is listed. Edits to manuscript have been highlighted in red where applicable.

Comments From the Editor

Comment 1:

Response 1:

The document and figures have been updated.

Comment 2: 

We note that the grant information you provided in the ‘Funding Information’ and ‘Financial Disclosure’ sections do not match. When you resubmit, please ensure that you provide the correct grant numbers for the awards you received for your study in the ‘Funding Information’ section.

Response 2:

Funding from the Minderoo foundation is provided in the Funding Information section, but no award numbers were supplied for these grants. I have updated the Award Number section to read “n/a” in the submission portal.

Comment 3:

"Earthrise Media and JJ received funding from the Minderoo Foundation (https://minderoo.org/) for this work. The funders had no role in study design, data collection and analysis, decision to publish, or preparation of the manuscript."

Response 3:

We have removed the Minderoo Foundation acknowledgement from the manuscript. The funding statement is accurate and does not need to be amended, unless the editors find that more information about funding is required.

Lines Changed: Line 532

Comment 4:

PLOS requires an ORCID iD for the corresponding author in Editorial Manager on papers submitted after December 6th, 2016. Please ensure that you have an ORCID iD and that it is validated in Editorial Manager.

Response 4:

The corresponding author’s ORCID has been linked.

Comment 5:

We note that Figures 1,2,4,5,6,8,11 and 12 in your submission contain [map/satellite] images which may be copyrighted. All PLOS content is published under the Creative Commons Attribution License (CC BY 4.0)

Response 5:

We have revised the figures such that all information shown is licenseable under CC-BY 4.0. Descriptions of the updates are provided below, and use the figure names from the first version of the manuscript:

Figures 1 and 2: The maps are created with data permissible for publication. Data sources are now listed in the figure captions. Licenses are as follows:

GADM - https://gadm.org/license.html:

“Using the data to create maps for publishing of academic research articles is allowed. Thus you can use the maps you made with GADM data for figures in articles published by PLoS, Springer Nature, Elsevier, MDPI, etc. You are allowed (but not required) to publish these articles (and the maps they contain) under an open license such as CC-BY as is the case with PLoS journals and may be the case with other open access articles.”

Natural Earth - https://www.naturalearthdata.com/about/terms-of-use/:

“All versions of Natural Earth raster + vector map data found on this website are in the public domain. You may use the maps in any manner, including modifying the content and design, electronic dissemination, and offset printing. The primary authors, Tom Patterson and Nathaniel Vaughn Kelso, and all other contributors renounce all financial claim to the maps and invites you to use them for personal, educational, and commercial purposes. No permission is needed to use Natural Earth. Crediting the authors is unnecessary.”

Figure 4: The imagery in the figure has been removed.

Figure 5: The imagery from Google has been replaced with imagery from Sentinel-2. This data comes from the European Space Agency’s Copernicus program and is open for public distribution and reproduction. The Copernicus Sentinel Data Agreement permits free access and use.

“EU law grants free access to Copernicus Sentinel Data and Service Information for the purpose of the following use in so far as it is lawful:

(a) reproduction;

(b) distribution;

(c) communication to the public;

(d) adaptation, modification and combination with other data and information;

(e) any combination of points (a) to (d).”

Figure 6: The imagery in this figure has been removed.

Figure 8: The high resolution imagery has been updated to use data from the Maxar Open Data program. This data has a CC BY-NC 4.0 license as follows

https://maxar-marketing.s3.amazonaws.com/files/downloads/119757_opendataprotocol_2020_04.pdf

For these open data, the event imagery and data layers will have a Creative Commons Attribution-NonCommercial 4.0 license (CC BY-NC 4.0). The conditions with that license mean that users are free to share, copy and redistribute the data in any medium or format under the following terms:

Attribution – you must give appropriate credit, provide a link to the license, and indicate if changes were made.

Non Commercial – you may not use the material for commercial purposes

Figures 11 and 12: These figures use data from Sentinel-2. See licensing information from figure 5.

Lines Changed:

The figure numbering has changed because of the paper restructuring and removal of figure 4. Mapping of figure numbers between revisions is shown below:

Fig 1 is now Fig 8

Fig 2 is now Fig 9

Fig 3 is now Fig 10

Fig 4 has been removed

Fig 5 is now Fig 11

Fig 6 is now Fig 12

Fig 7 is now Fig 1

Fig 8 is now Fig 2

Fig 9 is now Fig 3

Fig 10 is now Fig 4

Fig 11 is now Fig 5

Fig 12 is now Fig 6

Fig 13 is now Fig 7

Sup Fig 14 is now Sup Fig 13

Comment 6:

Please review your reference list to ensure that it is complete and correct. If you have cited papers that have been retracted, please include the rationale for doing so in the manuscript text, or remove these references and replace them with relevant current references.

Response 6:

The reference list is complete. No retracted papers have been cited. References have been added at the requests of the editor and reviewers. These are addressed in the responses to reviewer comments.

Lines changed: References # 15, 26, 35, and 40 have been added

Comments From Reviewer #1

Comment 1:

Introduction section included methods therefore, it might be useful to present a flowchart showing the steps involved.

Response 1:

We have restructured the paper such that the methods, including a methodological flow chart, now immediately follow the introduction.

Comment 2: 

Line 31, The manuscript includes a comprehensive literature review. I would advise referring to similar work funded by the JMSS featured here: https://www.undp.org/vietnam/blog/remote-sensing-smart-city-solution-municipal-waste-management

Response 2:

This work seems to be highly relevant to our study. Given that there are no peer-reviewed methods findings, it is difficult to assess this work in detail. However, we have included a note about the work in the introduction of the paper.

Changes:

… and a proof-of-concept system to detect municipal waste in Da Nang, Vietnam was demonstrated by the Japan Manned Space Systems Corporation and the Da Nang Institute for Socio-Economic Development [26].

Lines Changed: Line 42, reference #26

Comment 3:

Line 77: The interchangeable use of plastic and waste seems confusing. How do we know it is only plastics? The landfill sites consist of mixed waste, in most of developing countries, waste segregation is still a challenge. It might be useful to highlight the heterogeneity in the waste composition and discuss the percentage of plastics present across the geographical regions.

Response 3:

This is a helpful point of clarification. We have made it clearer that this system is created to identify municipal solid waste (MSW), defined as “discards from residential and commercial sources” (US EPA), and more clearly specified that detections are waste sites generally, rather than plastic waste sites specifically. 

We note the connection between plastic and MSW in lines 20-22, citing that 12% of MSW is plastic by mass globally. We did not quote a specific regional statistic since the study cited here estimates that plastic fraction in the East Asia and Pacific region is the same as the global average.

Changes:

In total, the model detected 374 waste aggregation sites across Indonesia (Fig 8) that trained reviewers were able to confirm through a manual review process that is detailed in the output validation section.

As another point of reference, we see that the number of waste sites detected…

Lines Changed: Lines 328 and 340

Comment 4:

Line 175: What do you mean by “South Asian biome.” I am not aware of any such biome. South Asia is a geographical construct.

Response 4:

We agree that characterizing this region as a biome is not correct, and have clarified the language in this sentence.

Changes:

The neural networks also show a capacity for geographic robustness, functioning in unseen countries across the geographically varied Southeast Asian region without additional tuning (Table 3).

Lines Changed: Line 447

Comment 5:

Line 453. Google and Mapbox host the base maps. Please provide the data description and resolution.

Response 5:

We have included a more detailed explanation of the imagery resolution and sources.

Changes:

Predominantly, validation is done through analysis of very-high resolution satellite data hosted on the Google Earth and Mapbox platforms. This imagery is a composite collected from multiple sensing platforms, but Very-High-Resolution imagery (ranging from under 60 cm and up to 5 m in resolution) is available in most cases [42].

Lines Changed: Line 302

Comment 6:

The discussion section is very weak. It should highlight future work and ways the results could be further improved.

Response 6:

We have updated the discussion to highlight limitations of the current model, the sources of bias that can enter through the validation procedures, and make note of what will be required in order to expand the system to operate globally.

Changes:

As discussed in the methodology, the validation of this data product relies on high-resolution satellite or aerial imagery. As such, detections present in the curated dataset are bounded by the quality and recency of the validation data source. In practice, this means that detected waste sites that appear after high-resolution validation data was collected will be rejected in the human validation stage. Additionally, validators may also improperly reject detected waste site candidates that are small, visually indistinct, or fall within areas where publicly-accessible validation data is of low quality. If desired, the Sentinel-2 detection system described in this work could inform a “tip and cue” system. Here, waste sites identified by the models could be added to a set of targets for high-resolution data collection or in-situ validation.

Though we have been able to develop models of impressive capability given limited labeled data, the detection system is not guaranteed to perform outside of the distribution of the data used to train the models. Since the training data comes from only Southeast Asia, we anticipate that more training data will be needed to identify waste sites in regions with ecosystems that are dissimilar to those in the domain of the current training data.

In future work, we will run this system globally. In order to accommodate this geographic expansion, we will continue to use the same data engine collection procedure for the new regions. As the domains of evaluation grow, the model sizes may need to increase as well in order to capture the additional training data variance.

Lines changed: Paragraphs starting at lines 430 and 471

Comment 7:

The paper doesn’t seem to follow the PLOS ONE format. Please refer to the journal outline and organize the sections accordingly. This will help improve the readability of the paper

Response 7:

We have restructured the paper such that the sections now follow a more traditional order. The methods now follow the introduction, instead of coming at the end of the paper.

Comments From Reviewer #2

Comment 1:

Abstract - Given the focal area of South East Asia, prominently describing this as working at continental scales seems slightly out of place. Maybe include that this can be expanded to continental scales towards the end of the abstract instead?

Response 1:

We have updated the abstract to better reflect the scale of analysis in this paper. The suggested revision is a more accurate depiction of the work detailed in this paper. We are optimistic about the system’s suitability to operate at continental scales. Since authoring this manuscript, the system has been used to find waste sites in more than 100 countries. 

Changes:

The system works at wide geographic scale, finding waste sites in twelve countries across Southeast Asia.

Lines Changed: Abstract

Comment 2: 

Line 3 - While you do go into more detail about the impacts of plastic waste, a good cite early on for “via pathways that are not fully understood” would help build a foundation of where this work fits into both the broader literature and real world applications.

Response 2:

This is a valuable clarifying point. We have removed the line about “pathways that are not fully understood,” and added more context showing how this work to directly monitor waste aggregations fits within the problem of plastic pollution.

Changes:

Since the sources of environmental plastics are broadly distributed geographically, most studies of pollution pathways have either focused on a selected set of field locations [13], or infer plastic waste density through modeling [14] [15].

Lines Changed: Line 10

Comment 3:

Line 69 - It seems like the paper was misformatted as the Results (and Discussion) sections come before the Methods (and data).

Response 3:

We have restructured the paper and the sections now follow a more traditional order. The methods now follow the introduction, instead of coming at the end of the paper.

Comment 4:

Table 1 - The amount of new detections and related rates are heavily influenced by Java. While this is just a piece of the results, I think it would be good to have some discussion of whether this is driven by island size, sample selection, location of actual waste, etc.

Response 4:

We believe that the number of detections in Java is driven by two factors. We ran the system in a high sensitivity configuration (see methods sections 3.2 and 3.3 and table 1 (table 4 in the previous manuscript). This mode is more likely to generate more waste sites at the cost of a greater false positive rate. We also think that the number of waste sites in Java is reasonable as Java contains a substantial fraction of the population of Indonesia. To show this correlation, we have updated the results tables (tables 2 and 3 in the latest manuscript) to include the population of each island and each country where we have detected waste sites. We have also added commentary about the relationship between population and number of detections in the results as well.

Changes:

As another point of reference, we see that the number of waste sites detected correlates well with the population within the region of evaluation (Pearson correlation, 0.991).

Once again, the number of detected waste sites within a country correlates well with its population (Pearson correlation, 0.960). Though this relationship is influenced by the country’s waste management practices, it provides grounding that the results are sensible in the absence of ground truth information on waste site distribution.

Lines Changed: See population column in tables 2 and 3, and lines 340 and 351

Comment 5:

Line 285 - What is the actual size (km) of Sentinel 2 patches around the centroids?

Response 5:

The patches are 480 meters in length. This information has been added to the text.

Changes:

For each labeled site location, we extract all 2019 Sentinel-2 L1C (top-of-atmosphere) data in a 480 × 480 meter square patch around the site centroid…

Lines Changed: Line 116

Comment 6:

Line 303 - How many iterations of the data engine did this go through? While this is iterative / ongoing for production, I assume there was a “final” number of iterations used for the results in this paper.

Response 6:

The final dataset is made up of thirteen separately-collected datasets produced through a bootstrapped sampling process. We have added this information to the data labeling section.

Changes:

After thirteen rounds of bootstrapping, the training dataset is made up of 213 locations containing waste and 345 regions without.

Lines changed: Line 112

Comment 7:

Line 378 - A cite for the Bayesian updates would be helpful.

Response 7:

We have not seen other work generate weak labels by combining classifier outputs using Bayesian updating. However, we have included a reference and note that this approach is within the domain of programmatic weak supervision.

Changes:

This technique is closely related to programmatic weak supervision labeling approaches [41].

Lines Changed: Line 209, reference #40

Comment 8:

Table 4 - Can you add more details on how you came up with the settings for the sensitivity modes?

Response 8:

We have included a more thorough description and grounding of these sensitivity parameters in the text.

Changes:

The sensitivity of the candidate detection stage can be tuned by controlling the prediction threshold and the minimum sigma (min sigma) value of the DoH algorithm. Tuning these parameters is a tradeoff between the precision and recall of detections. In absence of a database of known waste locations, we were not able to empirically sweep these values to find an optimum. Instead, we developed three sensitivity regimes detailed in Table 1 based on evaluations of candidates surfaced during model detection runs.

A higher classifier threshold leaves only regions which the networks assess as more certain to contain waste. The min sigma parameter is used by the DoH blob detection and controls the minimum standard deviation of the Gaussian blur kernel in the algorithm. As such, a lower min sigma parameter enables smaller and fainter blobs to be returned as candidates.

To put these values in context, a 3 × 3 pixel region with predicted values of 0.7 would be detected as a candidate using a min sigma value of 3.5, but would not be identified using a min sigma value of 5.0. Similarly, a 2 × 3 pixel region with predictions of 0.7 would not be surfaced at either min sigma value. If this 2 × 3 region had predicted values of 0.8, it would be identified using a min sigma of 3.5. The interplay between classifier thresholds and min sigma values are shown in fig. 7.

Lines Changed: Line 234

Comment 9:

The GitHub repository does not seem to be public at the moment. Was this kept private because the paper is under review?

Response 9:

The GitHub repository had been kept private while the paper was in initial review. The repository is now open and can be found at https://github.com/earthrise-media/plastics

---

## [Decision Letter · Decision Letter 1]

29 Nov 2022

Satellite monitoring of terrestrial plastic waste

PONE-D-22-22011R1

Dear Dr. Laurier,

We’re pleased to inform you that your manuscript has been judged scientifically suitable for publication and will be formally accepted for publication once it meets all outstanding technical requirements.

Kind regards,

Bijeesh Kozhikkodan Veettil

Academic Editor

PLOS ONE

Additional Editor Comments (optional):

Reviewers' comments:

Reviewer's Responses to Questions

**Comments to the Author**

1. If the authors have adequately addressed your comments raised in a previous round of review and you feel that this manuscript is now acceptable for publication, you may indicate that here to bypass the “Comments to the Author” section, enter your conflict of interest statement in the “Confidential to Editor” section, and submit your "Accept" recommendation.

Reviewer #2: All comments have been addressed

2. Is the manuscript technically sound, and do the data support the conclusions?

Reviewer #2: Yes

3. Has the statistical analysis been performed appropriately and rigorously? 

Reviewer #2: Yes

4. Have the authors made all data underlying the findings in their manuscript fully available?

Reviewer #2: Yes

5. Is the manuscript presented in an intelligible fashion and written in standard English?

Reviewer #2: Yes

6. Review Comments to the Author

Reviewer #2: All of my comments/concerns have been thoroughly addressed. I think this paper is in great shape and will be well received by readers.

7. PLOS authors have the option to publish the peer review history of their article (what does this mean?). If published, this will include your full peer review and any attached files.

Reviewer #2: No

---

## [Editor Report · Acceptance letter]

2 Dec 2022

PONE-D-22-22011R1 

Satellite monitoring of terrestrial plastic waste 

Dear Dr. Laurier:

I'm pleased to inform you that your manuscript has been deemed suitable for publication in PLOS ONE. Congratulations! Your manuscript is now with our production department. 

Kind regards, 

on behalf of

Dr. Bijeesh Kozhikkodan Veettil 

Academic Editor

PLOS ONE